# Beyond the Cup: Coffee Extracts as Modulators of Periodontal Inflammation and Bone Remodeling

**DOI:** 10.3390/cimb47100827

**Published:** 2025-10-08

**Authors:** Janvi Mody, Deamah Aleisa, Harshal Modh, Purnima Sainani, Serge Dibart, Weiyuan Ma

**Affiliations:** 1Department of Periodontology, Boston University Henry M. Goldman School of Dental Medicine, Boston, MA 02118, USA; daleisa@bu.edu (D.A.); sdibart@bu.edu (S.D.); weiyuanm@bu.edu (W.M.); 2Department of Preventive Dental Sciences, College of Dentistry, Imam Abdulrahman Bin Faisal University, Dammam 34212, Saudi Arabia; 3Department of Restorative Sciences and Biomaterials, Boston University Henry M. Goldman School of Dental Medicine, Boston, MA 02118, USA; hmodh@bu.edu; 4Henry M. Goldman School of Dental Medicine, Boston University, Boston, MA 02118, USA; purnima@bu.edu

**Keywords:** coffee, caffeine, chlorogenic acid, coffee polyphenols, alveolar bone loss, alveolar bone regeneration, inflammation, periodontitis

## Abstract

Alveolar bone loss is a defining feature of periodontitis and a principal cause of tooth loss worldwide. Driven by a dysregulated host immune response to chronic bacterial infection, periodontitis initiates a cascade of inflammatory events that lead to an imbalance in bone remodeling, favoring osteoclastic activity. While conventional periodontal therapies aim to control infection and inflammation, they often fall short in preserving bone integrity. As a result, interest has grown in adjunctive strategies targeting molecular pathways involved in bone metabolism. Among potential candidates, coffee, a globally consumed beverage often perceived as detrimental to health, has gained attention for its complex array of bioactive compounds, including caffeine, chlorogenic acids, and polyphenols. These compounds have demonstrated anti-inflammatory, antioxidant, and osteo-modulatory effects in various biological contexts. Despite coffee’s reputation as a potential health risk, its complex composition presents a paradox, necessitating an investigation into how its bioactive constituents may mitigate periodontal tissue destruction. The novelty of this short review lies in its integration of in vitro, animal, and epidemiologic evidence to delineate the dose- and context-dependent effects of coffee polyphenols, particularly chlorogenic and ferulic acids, on periodontal inflammation and alveolar bone remodeling, with special emphasis on osteoclast-related mechanisms that have not been synthesized previously. Caffeine can influence osteoblast and osteoclast activity in a dose-dependent manner, while chlorogenic acids (CGA) and polyphenols exert radical-scavenging and cytokine-suppressing activity that may reduce inflammatory bone loss. However, their efficacy is influenced by coffee species, cultivation, roasting, and extraction methods. This review evaluates current evidence and proposes directions for optimizing coffee-based formulations to support alveolar bone preservation in periodontitis.

## 1. Introduction

Periodontitis is a chronic inflammatory condition initiated by a complex interaction between a dysbiotic microbial biofilm and a dysregulated host immune–inflammatory response in genetically susceptible individuals, leading to the breakdown of the periodontal ligament and alveolar bone, ultimately compromising the structural support of the teeth [1,2,3]. This pathological process leads to the breakdown of the periodontal ligament and alveolar bone, ultimately compromising the structural support of the teeth [4]. Despite advancements in preventive care, the disease remains widespread, affecting nearly 50% of U.S. adults and over a billion people globally in its severe form [2,3]. This prevalence makes it one of the most significant non-communicable diseases worldwide and a major contributor to tooth loss, which can in turn elevate the risk for other systemic non-communicable diseases [5].

The underlying mechanisms of periodontitis are complex, driven by a persistent host immune response to the accumulation of microbial biofilm [6]. This dysregulated inflammation disrupts normal bone remodeling, tipping the balance toward bone resorption [2]. Elevated levels of inflammatory mediators such as interleukin-1β (IL-1β), tumor necrosis factor-alpha (TNF-α), and prostaglandin E2 (PGE2) stimulate osteoclast activity while impairing osteoblast function [7]. While conventional non-surgical and surgical therapies aim to halt the disease’s progression, the resulting alveolar bone loss is largely irreversible without advanced regenerative interventions. Therefore, there is growing interest in adjunctive strategies that can modulate underlying inflammatory pathways to better preserve existing bone structures.

In recent decades, regenerative dentistry has gained significant attention, with various techniques proposed to restore lost tissue. Among these, the application of different classes of biomaterials, each with specific chemical and biological features, has been central to promoting the regeneration of bone and soft tissues around teeth. Concurrently, there has been a shift toward exploring naturally derived compounds with anti-inflammatory and bone-protective properties [8].

Coffee, though often viewed through the lens of systemic health concerns, contains a range of bioactive compounds—including caffeine, chlorogenic acids, and polyphenols—that have shown promise in modulating inflammation and bone metabolism in other disease contexts [9]. At the molecular level, coffee’s bioactive constituents can modulate inflammatory pathways and bone cell activity relevant to periodontitis. For instance, caffeine has notable immunomodulatory effects: it can suppress neutrophil and monocyte chemotaxis as well as the production of pro-inflammatory cytokines like TNF-α and various T-cell cytokines. Such actions could hypothetically dampen the chronic periodontal inflammatory response that drives bone resorption [10]. Coffee polyphenols, especially chlorogenic acid, also exhibit potent antioxidant and anti-inflammatory properties [11]. In cell and animal models, chlorogenic acid has been shown to scavenge reactive oxygen species and reduce the expression of inflammatory mediators [10,12]. These effects are associated with activation of protective pathways (e.g., AMP-activated protein kinase and Nrf2/HO-1) and inhibition of NF-κB signaling, which together can mitigate inflammation-induced tissue damage [12].

Despite these insights, the role of coffee in periodontal inflammation and bone loss remains far from fully understood [13]. The existing literature presents a paradox: some epidemiological studies associate coffee consumption with increased periodontitis risk [14], while others find neutral or even protective effects [15]. This inconsistency likely stems from differing study designs, population factors, and the complex composition of coffee [13]. Crucially, there is a clear research gap regarding how coffee’s molecular actions translate to clinical outcomes in periodontitis. To date, few studies have directly examined coffee or its components in the context of inflammation-driven periodontal bone remodeling, and it remains uncertain whether coffee’s net effect is ultimately beneficial or harmful for alveolar bone preservation [13,15].

This review focuses on current evidence surrounding the effects of coffee extracts and their active constituents on inflammation-driven alveolar bone remodeling in periodontitis. We also consider how differences in coffee preparation methods may influence their therapeutic relevance, with the goal of identifying formulations that could serve as adjunctive tools in periodontal care.

## 2. Materials and Methods

### Search Strategy

A literature search was conducted using the PubMed, Scopus, and Google Scholar databases for articles published up to July 2025. The search strategy employed a combination of keywords, including “coffee,” “caffeine,” “chlorogenic acid,” “periodontitis,” “alveolar bone loss,” “inflammation,” “osteoblast,” and “osteoclast.” Both preclinical and clinical studies, including in vitro experiments, animal models, observational studies, and systematic reviews, were considered to provide a comprehensive overview of the topic. Databases searched: PubMed, Scopus, and Google Scholar; Period covered: January 2000–July 2025; Language: English; Article types included: in vitro, animal, and human (observational/interventional) studies relevant to coffee, caffeine, chlorogenic acid, periodontitis, alveolar bone loss, inflammation, osteoblasts, and osteoclasts. We conducted a review (PRISMA-ScR) and did not apply exclusion criteria for our short review. The figure found in the article was created by the authors using Notes application on iOs.

## 3. Periodontal Inflammation and Alveolar Bone Loss

Bone remodeling (or bone coupling) is a continuous process in which old bone is resorbed by osteoclasts and new bone is formed by osteoblasts, maintaining skeletal growth and structural integrity [16]. Under healthy conditions this process remains in balance, with no net change in bone mass [17]. In periodontitis, however, chronic inflammation skews this balance toward bone resorption, resulting in net loss of alveolar bone. The persistent inflammatory microenvironment, driven by a dysbiotic microbial biofilm in which keystone pathogens like *Porphyromonas gingivalis* play a critical role, not only causes direct tissue damage but also disrupts bone homeostasis by promoting osteoclastogenesis and suppressing osteoblast activity [18]. Elevated levels of Receptor Activator of NF-κB Ligand (RANKL)—induced by pro-inflammatory cytokines like IL-1β and TNF-α in inflamed periodontal tissues—excessively drive osteoclast differentiation and activation, tipping the remodeling equilibrium toward bone resorption [19]. Figure 1 illustrates the pathway of alveolar bone loss through inflammatory cytokines.

At the same time, *P. gingivalis* and its virulence factors impair osteoblast differentiation via TLR2-mediated signaling (e.g., through its atypical LPS and fimbriae) [20] and by inducing oxidative stress in local cells [21], thereby reducing new bone formation and the regenerative capacity of the periodontium. The bacterium’s cysteine proteases, gingipains, further exacerbate the imbalance by proteolytically degrading osteoprotegerin (OPG)—a decoy receptor that normally inhibits RANKL–RANK binding—effectively removing a critical checkpoint on osteoclast activation [22].

This dual mechanism of accelerated bone resorption coupled with suppressed bone deposition underlies the progressive alveolar bone loss characteristic of chronic periodontitis. As the bone is degraded and periodontal pockets deepen, anaerobic organisms like *P. gingivalis* gain access to newly exposed niches below the gum line, which reinforces microbial dysbiosis and sustains chronic inflammation in a self-perpetuating cycle. This vicious cycle ultimately destabilizes the tooth’s structural support (leading to loosening and, in severe cases, tooth loss) [23]. It also complicates therapy—the significant and often irreversible tissue destruction characteristic of advanced stages of periodontitis (e.g., Stage III and IV) is challenging to manage, making prevention a key clinical goal [24]. These insights underscore the importance of early intervention to halt inflammation before extensive bone damage occurs [23].

In addition to conventional mechanical debridement, adjunctive therapies targeting both the microbial triggers and the host’s bone remodeling response are being explored—for example, local or systemic antimicrobials/probiotics to rebalance the biofilm, and host-modulatory agents (like anti-cytokine therapies, antioxidants, or RANKL inhibitors) to curb inflammation-induced bone resorption [25]. Such combined strategies aim to break the destructive feed-forward loop in periodontitis, improving clinical outcomes and preserving alveolar bone health [23].

## 4. Bioactive Compounds in Coffee Extracts

There is a diverse array of bioactive compounds that make up coffee, many of which are believed to contribute to its potential therapeutic effects. The complex phytochemical profile of coffee includes caffeine, polyphenols (especially chlorogenic acids), diterpenoids (cafestol and kahweol), melanoidins, and trigonelline, each contributing uniquely to coffee’s antioxidant, anti-inflammatory, and neuroprotective properties [26].

Caffeine, a trimethylxanthine purine alkaloid, is perhaps the most recognized and studied compound in coffee [27]. It acts as a central nervous system stimulant, improving alertness and reducing fatigue, but also has important antioxidant and anti-inflammatory properties [28,29]. Caffeine can scavenge reactive oxygen species (ROS), regulate oxidative stress-related enzymes, and suppress inflammatory cytokines such as interleukin-6 (IL-6) and tumor necrosis factor-alpha (TNF-α). Through the modulation of microglial activity and adenosine A2 receptors, caffeine exerts neuroprotective and immunomodulatory effects, mechanisms which may also attenuate periodontal inflammation [30].

Polyphenols, particularly CGAs, are the most abundant phenolic compound in coffee, comprising up to 7–9% of the dry weight of green coffee beans [26,31]. The most prominent CGA is 5-caffeoylquinic acid (5-CQA), though feruloylquinic and dicaffeoylquinic acids are also present [29,31]. These compounds are powerful antioxidants, which directly neutralize ROS and upregulate the Nrf2 pathway, activating endogenous antioxidant enzymes like superoxide dismutase and glutathione peroxidase [28,30]. CGAs also inhibit NADPH oxidase, a pro-oxidant enzyme involved in oxidative damage. CGAs may modulate multiple inflammatory and oxidative stress pathways including Nuclear Factor Kappa-B (NF-κB) and Mitogen-Activated Protein Kinase (MAPK), both of which are implicated in the pathogenesis of periodontal disease [30]. CGAs have also demonstrated the ability to reduce the protease activity of *Porphyromonas gingivalis*, a key periodontopathogen, underscoring their potential as a natural antimicrobial agent in oral health applications [11,32]. Additionally, CGAs support glutamate homeostasis through promotion of glutamate transporter expression and by antagonizing NMDA receptors, both of which are mechanisms that may prevent neuroinflammation and excitotoxicity [30].

Other phenolic acids in coffee include caffeic acid, ferulic acid, and protocatechuic acid [26]. Caffeic acid often presents as an ester with CGAs, but it also has independent anti-inflammatory effects by inhibiting the NF-κB and MAPK signaling and enhancing Nrf2-mediated antioxidant enzyme expression [27,30]. These mechanisms may translate to reduced inflammatory responses with periodontal tissues. 

The diterpenoids cafestol and kahweol, present in the lipid fraction of coffee, have shown anticarcinogenic, antioxidant, and anti-inflammatory activity. These compounds include phase II detoxification enzymes, including glutathione S-transferase, and modulate xenobiotic metabolism in gut and oral mucosa. Kahweol, in particular, is more abundant in Arabica coffee, and its antioxidant properties may contribute to cellular resilience against oxidative damage in inflamed periodontal tissues [26].

Melanoidins are higher-molecular-weight, polymeric compounds formed during the Maillard reaction in the roasting process, comprising up to 25% of coffee’s dry matter [26,28,33]. They are structurally complex and can incorporate other phenolic molecules [28,33]. Melanoidins exhibit strong antioxidant properties via metal chelation and free radical scavenging and also have antimicrobial activity through cell membrane disruption and inhibition of bacterial siderophores [26]. Their contribution to periodontal health is promising, although their complex structure presents challenges in identifying their specific biological roles and absorption profiles. 

Trigonelline is a nitrogen-containing alkaloid that undergoes transformation into other beneficial compounds during roasting. It has been shown to have neuroprotective activity, potentially improving cognitive function and crossing the blood–brain barrier. Its ability to activate endogenous antioxidant pathways suggests that it could contribute to anti-inflammatory effects relevant in periodontitis, although further studies are needed to clarify its role [26]. A summary of all the key bioactive compounds of coffee can be appreciated in the table below (Table 1). 

**Table 1 cimb-47-00827-t001:** Summary of key bioactive compounds in coffee and their periodontal relevance.

Bioactive Compound	Primary Effects on Periodontal Health	Proposed Mechanism of Action	Key Takeaway for Periodontal Therapy
**Caffeine** [10,32,34,35]	Biphasic effects. High doses may exacerbate bone loss and impair healing. Low doses may have mild anti-inflammatory effects.	Adenosine receptor antagonism; suppression of cytokines like TNF-α.	A primary source of risk. Its removal or strict dose control in therapeutic formulations is critical.
**Chlorogenic Acids (CGA) *** [11,36,37]	Potent anti-inflammatory, antioxidant, and antimicrobial effects. Promotes osteoblast function and suppresses osteoclast activity.	Inhibits NF-κB and MAPK signaling; activates Nrf2 antioxidant pathway; reduces *P. gingivalis* protease activity.	mitigates periodontitis by reducing *P. gingivalis* virulence, suppressing oxidative stress and inflammatory signaling (NF-κB, MAPK), and restoring bone balance through enhanced osteoblast activity and inhibition of osteoclastogenesis
**Diterpenes** (Cafestol & Kahweol) [38]	Anti-inflammatory and antioxidant properties.	Modulate detoxification enzymes; enhance the Nrf2 antioxidant response.	Beneficial, but levels are heavily dependent on coffee species (higher in Arabica) and preparation (removed by paper filters).
**Melanoidins** [39,40]	Strong antioxidant and antimicrobial activity.	Formed during roasting via the Maillard reaction. Exhibit metal chelation and free radical scavenging properties.	Contribute to the overall antioxidant capacity of roasted coffee, but their complex structure makes specific roles hard to study.

* The most promising component for therapeutic use. Extracts should be optimized to maximize CGA content.

## 5. Variability in Composition: Source, Processing and Composition

The therapeutic potential of coffee is significantly influenced by its chemical composition, which varies based on species, geography, processing, and preparation methods. Coffea robusta beans generally contain higher levels of caffeine and CGAs than Coffea arabica, although Arabica contains more kahweol [29].

Roast level also plays a significant role; light roasts retain more polyphenols, while dark roasts though lower in certain heat-sensitive compounds, generate melanoidins that contribute to antioxidant activity [26]. Brewing methods influence extract composition as well; espresso typically yields higher concentrations of polyphenols and caffeine per unit volume than drip coffee, although differences in serving size may offset this effect [30].

Additionally, simulated gastrointestinal digestion has been shown to increase the bioavailability and activity of phenolic compounds, effectively doubling antioxidant potential in vitro [33]. Factors such as the addition of milk or sugar, consumption frequency, and individual metabolism also influence the biological effects of coffee [35]. Therefore, understanding these variables is essential when assessing coffee’s impact on oral and systemic health.

## 6. Properties of Coffee

### 6.1. Anti-Inflammatory Properties

Coffee polyphenols, especially CGAs and caffeic acid, have been shown to inhibit key inflammatory mediators involved in chronic diseases. They downregulate NF-κB and MAPK pro-inflammatory signaling pathways [30] and decrease levels of pro-inflammatory cytokines such as IL-6, TNF-α, and interleukin-1 beta (IL-1β) [41].

Caffeine suppresses cytokine production and modulates immune cell phenotypes, notably shifting microglial cells toward an anti-inflammatory M2 phenotype [42]. These effects mirror pathogenic processes of periodontal inflammation, where overproduction of IL-6 and TNF-α drives connective tissue degradation and osteoclastogenesis [30]. While most studies examining these mechanisms were conducted in the context of neuroinflammation or systemic diseases, the underlying pathways are also central to periodontitis pathophysiology, suggesting a theoretical basis for coffee extracts to modulate periodontal inflammation.

### 6.2. Antioxidant Effects

Periodontal disease is associated with elevated oxidative stress, which increases tissue destruction. Coffee constituents counteract this through both direct free radical scavenging and activation of endogenous defense systems [26,30].

CGA, ferulic acid, and caffeine demonstrate potent ROS-scavenging activity, reducing lipid peroxidation and DNA damage [26,28]. Additionally, coffee activates the Nrf2 antioxidant pathway, promoting expression of protective enzymes such as superoxide dismutase, glutathione peroxidase, and heme oxygenase-1 (HO-1) [30]. Diterpenes like kahweol further enhance these responses [28]. Notably, in vitro digestion of coffee increases its antioxidant capacity, enhancing its potential efficacy when consumed [30].

These antioxidant effects, although studied outside of the oral cavity, are likely beneficial in mitigating oxidative damage in periodontal tissues, offering indirect protection against inflammatory alveolar bone loss.

## 7. Identifying Optimal Coffee Components for Bone Health

### 7.1. Analysis of Which Coffee Components Most Effectively Support Periodontal Bone Health

Coffee has long been a complex beverage containing several bioactive components like caffeine, a variety of polyphenols such as CGA, and minerals like potassium all of which can influence bone metabolism [43].

Caffeine, which is one of coffee’s main ingredients, has a dose-dependent impact on bone health. High caffeine intake may inhibit bone formation and promote calcium loss (through mechanisms like adenosine receptor antagonism and altered vitamin D/calcium homeostasis), potentially leading to reduced bone density [44]. In contrasting nature, coffee’s rich polyphenols exhibit antioxidant and anti-inflammatory properties that can protect bone tissue from oxidative stress and inflammation-related damage [12]. Different types of coffee vary significantly in their bioactive compound content and potential bone-related effects as summarized in Table 2.

Among these, CGA, which is a major coffee polyphenol being especially notable for its osteogenic effects this CGA promotes the survival, differentiation, and activity of bone-forming osteoblast cells while suppressing the development and function of bone-resorbing osteoclasts [45]. Consistently, animal studies have demonstrated that CGA supplementation can prevent bone loss; for example, chlorogenic acid administration in estrogen-deficient (ovariectomized) rats enhanced osteoblast function, inhibited osteoclastogenesis, and ultimately preserved bone mass [9].

Taken together, the net effect of coffee on skeletal health reports to depend on the balance of its components: while antioxidant polyphenols like chlorogenic acid may show osteoprotective benefits, high caffeine could offset these gains, underscoring the importance of optimizing coffee’s component mix for bone health [43].

**Table 2 cimb-47-00827-t002:** Comparison of coffee drinks based on bioactive content and potential bone health benefits.

Coffee Type	Caffeine Content	Chlorogenic Acids	Polyphenols	Diterpenes	Antioxidant/Bone Implication
Espresso	High per mL	High	Moderate to High	High (unfiltered)	Potent antioxidants may support bone health if intake is moderate [46]
Drip Coffee	Moderate	Moderate to High	High	Low (filtered)	Suppresses osteoclastogenesis; favorable for bone [47]
Instant Coffee	High (varies by brand)	Moderate (preserved by freeze-drying)	Moderate	Low	Offers antioxidant support; lower diterpenes; caffeine impact depends on dose [48]

### 7.2. Impact of Roasting Levels and Brewing Methods on Coffee’s Bioactive Components

Roast degree significantly alters coffee’s phytochemical profile for example, light roasted beans tend to retain much higher levels of chlorogenic acids and total polyphenols than dark-roasted beans, since prolonged high heat degrades these antioxidant compounds [49]. Similarly, the preparation method influences component efficacy: different brewing techniques yield varying phenolic contents and antioxidant capacities in the final coffee beverage [50].

These compositional differences can translate into functional effects; indeed, A study showed that the antioxidant and anti-inflammatory benefits of coffee varied depending on the roasting level used [51].

Much of coffee’s potential in periodontal therapy is attributed to its polyphenols (especially chlorogenic acid), which exhibit notable antibacterial and anti-inflammatory activities [11]. In fact, coffee extracts rich in chlorogenic acid such as those from green (lightly roasted or unroasted) beans have demonstrated potent antimicrobial effects against periodontal pathogens and even the ability to attenuate inflammation and alveolar bone loss in periodontitis models [11,52].

These findings underscore the importance of standardizing roast level and brewing conditions when developing coffee-based extracts for periodontal therapy, to ensure consistently high levels of the key active components and reproducible therapeutic efficacy [53].

In vitro studies have highlighted the diverse biological activities of coffee and its bioactive compounds in the context of periodontal health. Coffee extract and chlorogenic acid were shown to inhibit *P. gingivalis* growth and gingipain activity [11], while robusta coffee extract demonstrated antibacterial effects against periopathogens, attributed to caffeine, trigonelline, flavonoids, and chlorogenic acid [54]. In addition to antimicrobial actions, coffee-derived compounds exhibited anti-inflammatory and antioxidant properties: coffee extract reduced pro-inflammatory cytokines (IL-6, IL-8) in LPS-stimulated oral keratinocytes [10], and coffee phenolics activated the Nrf2/HO-1 antioxidant pathway while reducing ROS in endothelial cells [55]. However, not all effects were protective, as caffeine enhanced RANKL-mediated osteoclastogenesis in compressed human PDL cells [56], suggesting a possible contribution to bone resorption under mechanical stress. Collectively, these findings indicate that coffee and its constituents exert multifaceted actions which are antimicrobial, anti-inflammatory, antioxidant, and pro-osteoclastogenic properties reflecting their complex role in periodontal tissue biology (Table 3).

**Table 3 cimb-47-00827-t003:** Summaries of in vitro studies have examined the effects of coffee and its compounds on periodontal pathogens and host responses.

Study Type	Reference	Model & Sample Size	Primary Outcomes & Findings
**In vitro**	Tsou, 2019 [11]	*P. gingivalis* cultures	Coffee extract & chlorogenic acid inhibited bacterial growth and gingipain activity
	Sari, 2023 [54]	Robusta coffee extract vs. periopathogens	Antibacterial activity linked to caffeine, trigonelline, flavonoids, chlorogenic acid
	Song, 2022 [10]	Human oral keratinocytes (LPS-stimulated)	Coffee extract reduced pro-inflammatory cytokines (IL-6, IL-8)
	Lonati, 2022 [55]	Cell stress models (endothelial cells)	Coffee phenolics activated Nrf2/HO-1 antioxidant pathway and reduced ROS
	Yi, 2016 [56]	Human PDL cells under compression	Caffeine enhanced RANKL-mediated osteoclastogenesis

Evidence from animal models has provided important insights into how coffee and its major bioactive component, caffeine, may influence periodontal and alveolar bone biology. Several experimental studies in rats have demonstrated adverse effects of caffeine on bone healing and loss. Reference [57] reported that both coffee and caffeine delayed bone repair in extraction sockets, with caffeine exerting the stronger inhibitory effect. Similarly, [58] showed that high-dose caffeine aggravated alveolar bone loss in ligature-induced periodontitis. In orthodontic models, caffeine accelerated tooth movement and induced alveolar bone loss through stimulation of osteoclastogenesis and altered RANK/RANKL/OPG signaling [56,59]. In contrast, other reports highlight more neutral or even protective effects. Reference [60] observed no significant bone loss under their dietary coffee regimen, and [61] reported mixed but largely neutral skeletal effects with moderate caffeine dosing. Notably, [12] demonstrated that coffee intake reduced oxidative stress and attenuated alveolar bone loss in aged rats. These animal study findings indicate mixed evidence, while high doses of caffeine and inflammatory settings consistently show detrimental effects on bone, moderate coffee intake or whole-coffee exposure may be neutral or even beneficial, particularly through antioxidant pathways (Table 4).

**Table 4 cimb-47-00827-t004:** Summaries of in vivo studies have examined the effects of coffee and its compounds on periodontal pathogens and host responses.

Study Type	Reference	Model & Sample Size	Primary Outcomes & Findings
**In vivo**	Macedo, 2015 [57]	Rat extraction socket bone healing	Coffee & caffeine delayed alveolar bone repair; caffeine stronger effect
	Bezerra, 2008 [58]	Ligature periodontitis in rats	High caffeine intake worsened alveolar bone loss
	Yi, 2016 [56]	Rat orthodontic tooth movement	Caffeine accelerated tooth movement via increased osteoclast activity
	Moreno, 2024 [59]	Rat orthodontic model	Caffeine induced alveolar bone loss via RANK/RANKL/OPG signaling during tooth movement
	Sakamoto, 2001 [60]	Rat diet/bone metabolism	Coffee intake did not induce bone loss under that experimental regime
	Folwarczna, 2017 [61]	Rat skeletal bone model	Moderate caffeine dosing had mixed/largely neutral effects on bone parameters
	Kobayashi, 2020 [12]	Aged rats, periodontal tissues	Coffee reduced oxidative stress and alveolar bone loss

## 8. Clinical Evidence and Human Studies in Periodontitis

Although preclinical studies suggest that coffee and its bioactive compounds particularly chlorogenic acids and caffeine may modulate inflammatory and oxidative pathways relevant to periodontal disease, clinical evidence in human populations remains inconsistent and limited. Observational studies have yielded mixed results, with some reporting potentially protective associations and others indicating increased risk of periodontitis with high coffee intake.

Data from the Hamburg City Health Study (*n* = 6209) demonstrated a positive association between high coffee consumption (≥7 cups/day) and periodontitis prevalence, although moderate intake showed no such relationship [13]. Conversely, the VA Dental Longitudinal Study found that higher coffee consumption in older men was associated with reduced alveolar bone loss over time [62], suggesting a potential protective effect in specific populations. 

Further complexity arises from biomarker-based studies. A 2024 NHANES analysis identified a positive correlation between urinary caffeine metabolites such as 1-methyluric acid and 1,7-dimethylxanthine and periodontitis severity [63], implying a possible metabolic link. Similarly, a Mendelian randomization analysis found a weak but statistically significant causal relationship between genetically predicted coffee consumption and increased periodontitis risk, though the effect size was minimal [34]. Table 5 shows a detailed summary of all these studies.

**Table 5 cimb-47-00827-t005:** Summaries of human studies and clinical trials have examined the effects of coffee and its compounds on periodontal pathogens and host responses.

Study/Analysis	Design & Population	Findings
Hamburg City Health Study (*n* = 6209) [13]	Cross-sectional	High intake (≥7 cups/day) causes increased periodontitis prevalence; moderate intake shows no association
VA Dental Longitudinal Study [62]	Older men, prospective	Higher coffee consumption decreases alveolar bone loss over time
NHANES 2024 (biomarker-based) [63]	Urinary metabolites (1-methyluric acid, 1,7-dimethylxanthine)	Positive correlation with periodontitis severity
Mendelian randomization analysis [34]	Genetic epidemiology	Weak but significant causal link between genetically predicted coffee intake & increased periodontitis risk (minimal effect size)

Overall, these findings highlight a complex and context-dependent relationship between coffee consumption and periodontal health. Variations in coffee type (e.g., filtered vs. unfiltered, instant vs. brewed), preparation methods, additive use (e.g., sugar or milk), and individual metabolic responses may all contribute to the observed discrepancies. Importantly, there remains a lack of interventional or randomized controlled trials (RCTs) assessing the direct effect of standardized coffee extracts on clinical periodontal parameters such as probing depth, clinical attachment loss, or radiographic bone support.

## 9. Safety and Considerations for Periodontal Applications

The role of coffee and its bioactive compounds in the modulation of periodontitis-related bone remodeling is complex and multifaceted, underscoring the need for careful safety evaluations prior to therapeutic application. The dual nature of coffee’s profile lies in its composition with compounds such as caffeine, CGAs, melanoidins, and various polyphenols, as they can exert either protective or deleterious effects on alveolar bone and oral tissues depending on their concentration, delivery modality, and patient-specific variables [30].

On the protective end of the spectrum, many in vitro and in vivo studies have demonstrated that polyphenols found in coffee, particularly CGA and caffeic acid, exhibit strong antioxidant and anti-inflammatory effects that are advantageous in the context of periodontitis. These compounds have been shown to activate the Nrf2 signaling pathway, thereby enhancing systemic antioxidant defenses and mitigating the oxidative stress that contributes to alveolar bone degradation [12]. Animal studies lend further support to these findings; for instance, rodents subjected to chronic coffee intake exhibited the preservation of alveolar bone structure, with histological assessments revealing elevated osteoblast activity and increased expression of osteogenic markers such as bone morphogenic protein-2 (BMP-2). In addition to their bone-preserving effects, these polyphenols also have notable antimicrobial properties, particularly against periodontopathogenic bacteria like *Porphyromonas gingivalis*, suggesting a dual action mechanism where microbial dysbiosis is addressed alongside bone preservation [35].

However, these findings must be tempered with a clear understanding of the risk posed by certain coffee constituents, particularly caffeine. Studies have repeatedly shown that high-dose caffeine administration, particularly in the context of ligature-induced periodontitis models, can exacerbate bone loss and impair the healing process following extractions [35,57]. In some rodent models, caffeine exposure resulted in a 60% reduction in new bone formation compared to controls [57]. This raises serious concerns about the safety of caffeine-rich coffee extracts in periodontal therapy, especially when used above physiological concentrations. This also highlights the importance of rigorous dose–response studies to delineate therapeutic thresholds from toxic exposures. 

Beyond the impact on bone, the habitual consumption of coffee and its application in oral therapeutics carries additional implications for oral health. Tooth loss has been correlated with high coffee intake in population-based studies, a phenomenon largely attributed to the widespread consumption of sugar-sweetened instant coffee mixes. These additives promote cariogenic plaque development, accelerating enamel demineralization and ultimately contributing to tooth decay [64].

In contrast, unsweetened black coffee can exert mildly protective effects against caries due to its antibacterial polyphenols, although this benefit is contingent upon the absence of fermentable sugars [34,64]. The influence of caffeine on bone mineral density (BMD) is another area of concern, particularly for postmenopausal women [65]. It interferes with calcium absorption, suppressing vitamin D activity, and increasing urinary calcium excretion, thereby compromising systemic and oral bone health [35]. Nevertheless, moderate coffee consumption, defined as less than 400 mg of caffeine per day, appears to have a neutral or even slightly beneficial impact in some studies, including those involving ovariectomized rats, as the model for postmenopausal osteoporosis [9,65]. 

Other potential side effects include extrinsic dental staining and thermal dentin hypersensitivity. Tannins present in coffee are well-known contributors to enamel discoloration, a cosmetic issue often observed in clinical practice. While anecdotal reports link hot coffee consumption with increased dentin sensitivity, empirical evidence in this domain remains sparse, and further research is needed to evaluate whether these outcomes are enhanced by the applications of concentrated extracts in therapeutic settings.

Establishing a safe and effective dosing protocol is central to the therapeutic utility of coffee extracts in periodontics. Given the biphasic nature of many of its bioactive compounds, where low concentrations yield benefits and higher doses cause harm, precision in dosing is imperative. General safety guidelines suggest that up to 400 mg of caffeine per day is well-tolerated by healthy adults; however, this reference point pertains to systemic dietary intake and may not translate directly to localized periodontal applications within the periodontal environment, where pharmacokinetics and tissue exposure profiles differ substantially [65]. Preclinical studies suggest that a consistent intake of coffee containing approximately 1.36% active compound can yield bone-protective effects, whereas higher concentrations equivalent to daily consumption of 16 cups of coffee induce significant bone loss and cellular toxicity [12,35,62]. 

The mode of delivery also plays a critical role in modulating both efficacy and safety. The topical or localized application of Robusta coffee bean extract into periodontal pockets has shown promising results in reducing alveolar bone loss and microbial load in animal models. This targeted delivery approach offers the advantage of minimizing systemic exposure and allows for higher local concentrations of therapeutic compounds without crossing systemic toxicity thresholds [11,54]. In contrast, systemic delivery by oral consumption presents several challenges, including first-pass hepatic metabolism and inter-individual variability in absorption [57,62,64]. These factors reduce bioavailability and may necessitate higher doses for efficacy, thus increasing the risk of adverse outcomes. As such, topical delivery systems, such as rinses, gels, or biodegradable microspheres containing standardized coffee-derived polyphenols, may represent a safer and more effective route for clinical application [11,54,57,62,64].

The phytochemical profile of the coffee extract in question is another crucial determinant of its therapeutic potential. Not all coffee extracts are biochemically equivalent, and their composition is shaped by a host of variables including species, origin, roasting level, and brewing method [65]. CGAs, known for their antioxidant and antimicrobial effects, are more abundant in lightly roasted beans and tend to be higher in Robusta varieties compared to Arabica [29,30]. This compositional difference can influence extract selection depending on the clinical goal of prioritizing either antimicrobial action or bone preservation. While caffeine can exhibit mild anti-inflammatory effects at low doses, its detrimental influence on bone metabolism at higher concentrations makes it a compound that requires strict titration or removal [35].

Decaffeinated extracts may therefore offer a preferable alternative in therapeutic formulations. Melanoidins and diterpenes, such as kahweol and cafestol, formed during the roasting process, add further antioxidant capacity though their specific impact on periodontal health remains uncertain [33]. Other less-characterized components such as trigonelline, which is unknown for its neuroprotective effects, may also support systemic health, although the relevance to periodontal tissues has yet to be clearly defined [26]. Additionally, the inclusion of additives in therapeutic formulations must be carefully evaluated. While dairy components may synergize with bioactives to support remineralization, sugars and syrups introduce cariogenic risk and may negate otherwise beneficial effects [35,65].

## 10. Future Directions and Research Gaps

While it is well established that lipopolysaccharide (LPS) exposure activates key inflammatory signaling pathways such as NF-κB and MAPKs in osteoblasts, leading to increased cytokine production and subsequent bone resorption, the modulatory effects of coffee extracts on these pathways remain poorly characterized. Most existing studies have focused on individual bioactive compounds (e.g., caffeine, chlorogenic acid), rather than the full phytochemical complexity of whole-coffee extracts.

Moreover, the literature to date has disproportionately centered on osteoblasts, with limited exploration of how coffee components affect osteoclast differentiation, activation, and function under similar inflammatory conditions. This represents a critical knowledge gap, given the tightly coupled relationship between osteoblasts and osteoclasts in maintaining bone homeostasis. Although some polyphenols in coffee have demonstrated anti-inflammatory and antioxidant potential, their specific roles in modulating osteoclastogenesis particularly in LPS-driven inflammatory microenvironments remain largely uninvestigated.

While numerous studies highlight the antioxidant and anti-inflammatory potential of coffee constituents, it is equally important to recognize the adverse effects reported for caffeine. Experimental evidence demonstrates that high-dose caffeine intake can exacerbate alveolar bone loss and enhance osteoclast activity, particularly under inflammatory conditions. Epidemiological data have also associated excessive or sugar-sweetened coffee consumption with greater risk of periodontitis and tooth loss. These findings underscore the importance of dose moderation and formulation context when interpreting caffeine’s role in periodontal and bone health.

To advance this field, future research should focus on elucidating the **molecular effects of whole-coffee extracts** on both osteoblast and osteoclast activity during LPS-induced inflammation, capturing the dynamic interplay between bone-forming and bone-resorbing cells. It is equally important to **investigate potential synergistic or antagonistic interactions among coffee’s bioactive constituents**, moving beyond isolated compound studies to reflect the extract’s natural complexity. Further, clarifying the **bidirectional crosstalk** between osteoblasts and osteoclasts in the presence of coffee-derived molecules will provide new insights into how these constituents influence inflammatory bone remodeling. Finally, **determining whether coffee consumption constitutes a modifiable dietary factor** that affects the onset or progression of inflammation-driven bone loss, such as that observed in periodontitis, remains an important objective for translational research.

In parallel, addressing the clinical relevance of coffee in periodontal health will require rigorous human studies. Future investigations should prioritize **randomized controlled trials** assessing the therapeutic efficacy of standardized coffee extracts which are administered either systemically or locally, mainly on periodontal inflammatory markers, clinical parameters, and alveolar bone preservation. Complementary **prospective cohort studies** are needed to control for confounding factors such as smoking, oral hygiene, systemic conditions, and dietary habits, while accounting for variations in coffee type, preparation method, and consumption frequency, including the addition of milk and sugar. **Dose–response analyses** will be essential to establish thresholds at which coffee-derived components (e.g., caffeine, chlorogenic acid) exert protective versus detrimental effects on periodontal tissues. **Mechanistic clinical studies incorporating molecular endpoints**—such as IL-1β, TNF-α, malondialdehyde, Nrf2 activation, and the RANKL/OPG ratio—could further clarify biochemical–clinical correlations. Finally, **evaluating novel delivery systems**, including topical formulations (e.g., gels, rinses, or microspheres) enriched with decaffeinated or polyphenol-rich coffee extracts, may offer promising localized strategies for adjunctive periodontal therapy.

Collectively, these directions can substantially deepen our understanding of coffee’s therapeutic potential in periodontal care, supporting the development of innovative interventions to mitigate inflammation-induced alveolar bone loss and promote oral health maintenance.

## 11. Conclusions

The evidence reviewed in this article highlights the nuanced role of coffee and its bioactive constituents in modulating inflammation, oxidative stress, and bone metabolism, key processes in the pathophysiology of periodontitis. While caffeine, at high doses, has been associated with impaired bone formation and calcium depletion, the polyphenolic compounds in coffee, especially chlorogenic acid, have demonstrated potent anti-inflammatory, antioxidant, and osteoprotective effects in both in vitro and in vivo models. These findings collectively suggest that certain components of coffee, when optimized for concentration and delivery, may offer therapeutic benefits in periodontal disease management, particularly by protecting against alveolar bone loss.

However, the functional impact of coffee on periodontal tissues appears highly dependent on its preparation method, roast level, and mode of administration. Lightly roasted and unfiltered coffee preparations retain higher levels of chlorogenic acids and polyphenols, enhancing their therapeutic potential. Conversely, high-temperature roasting and sweetened or highly caffeinated commercial coffee products may negate these benefits or introduce additional oral health risks. Importantly, emerging evidence suggests that topical or localized delivery systems like gels or microspheres infused with standardized, polyphenol-enriched coffee extracts may allow for targeted periodontal therapy without the systemic risks associated with high-dose caffeine consumption.

Despite promising biochemical and preclinical findings, significant research gaps persist. There is an urgent need for well-designed human studies and clinical trials to validate these therapeutic claims, establish safe dosing thresholds, and explore the mechanistic interactions between coffee constituents and bone cells within inflamed periodontal environments. The standardization of extract composition and delivery methods will be critical for ensuring consistent outcomes. With continued research, coffee-derived formulations may evolve from a dietary interest into a viable adjunct in periodontal care, offering an innovative, naturally sourced strategy to combat inflammation-driven bone loss. Therefore, future efforts should prioritize the development and clinical testing of decaffeinated, chlorogenic acid (CGA)-enriched topical formulations. This approach represents a promising and potentially safer therapeutic strategy to harness the osteoprotective benefits of coffee polyphenols while avoiding the well-documented risks associated with high concentrations of caffeine.

## Figures and Tables

**Figure 1 cimb-47-00827-f001:**
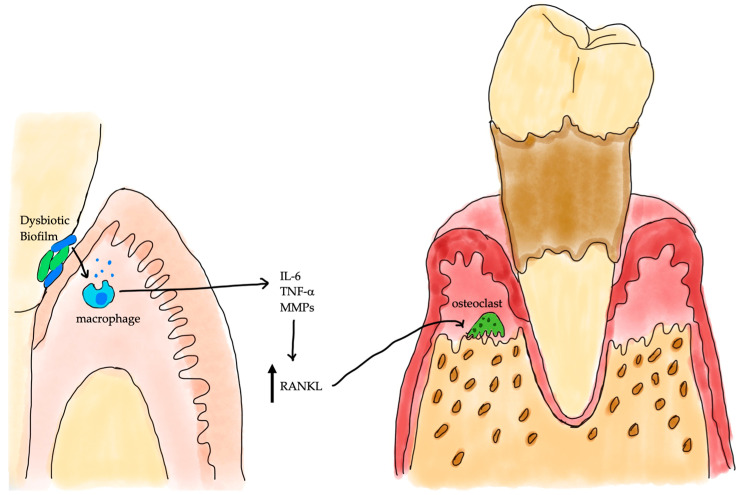
Schematic representation of the host–microbe interactions driving periodontal bone loss. A dysbiotic subgingival biofilm stimulates macrophages to release pro-inflammatory cytokines (IL-6, TNF-α) and matrix metalloproteinases (MMPs), which increase receptor activators of nuclear factor-κB ligand (RANKL) expression. Elevated RANKL promotes osteoclast differentiation and activity, resulting in alveolar bone resorption typical of periodontitis.

## Data Availability

No new data were created or analyzed in this study.

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
