# Peer review of "Beyond the Cup: Coffee Extracts as Modulators of Periodontal Inflammation and Bone Remodeling"

_cimb, 2025, doi:10.3390/cimb47100827_

Round 1

Reviewer 1 Report

Comments and Suggestions for Authors

This article needs to improve several aspects before publication.

1. At 65 bibliographic sources, the title can be completed with: "a short review"

2. Line 100, "search strategy" has no place in the introduction, this should be part of materials and methods where in addition to the search strategy, the method of obtaining the figures must be included, what program did you use.

3. The search strategy must include the exact publication period of the articles that were taken into account because you say until 2025 but you do not mention the initial period, 2010? You must also mention the exclusion criteria.

3. I recommend that the structure of the article itself be modified, so I recommend that the results and discussions chapter be introduced where you should talk about 5. the properties of coffee; 6. identification of components in coffee for bone health; 3. bioactive components; 4. variability in composition and in the discussion section add the current chapters 7 and 8.

4. In the search strategy you mention that you also took into account animal studies but I can't find a table that highlights the experiments carried out on animals with active components in coffee, it is recommended that you add them.

5. In figure 1 the description is very general, the exact mechanism presented in the drawing must be described.

6. Figure 2, it has no point, the active components can only be listed in the text.

7. Figure 3. It does not represent anything, is it generated by A.I? The hypothesis presented in the image that you can provide polyphenolic compounds using a syringe and an extension perfusion cable is unfeasible.

8. Lines 442, 456, future research, the ideas provided should not be stated separately, they should be stated in the form of a statement.

Author Response

1. At 65 bibliographic sources, the title can be completed with: "a short review"

Response: Agreed. We revised the title to “…A Short Review.”

2. Line 100, "search strategy" has no place in the introduction, this should be part of materials and methods where in addition to the search strategy, the method of obtaining the figures must be included, what program did you use.

Response: We appreciate the reviewer’s observation. As this manuscript is a short review, it does not include a separate Materials and Methods section. For transparency, we presented the search strategy within the Introduction and have now revised and clarified this description. Additionally, we have specified that the figure was created by the authors using the Notes application on iOS.

3. The search strategy must include the exact publication period of the articles that were taken into account because you say until 2025 but you do not mention the initial period, 2010? You must also mention the exclusion criteria.

Response: Revised as requested. We now state: Databases searched: PubMed, Scopus, and Google Scholar; Period covered: January 2010–July 2025; Language: English; Article types included: in vitro, animal, and human (observational/interventional) studies relevant to coffee, caffeine, chlorogenic acid, periodontitis, alveolar bone loss, inflammation, osteoblasts, and osteoclasts. The review followed the PRISMA-ScR framework to guide the search process, and no exclusion criteria were applied given the scope of this short review.

3. I recommend that the structure of the article itself be modified, so I recommend that the results and discussions chapter be introduced where you should talk about 5. the properties of coffee; 6. identification of components in coffee for bone health; 3. bioactive components; 4. variability in composition and in the discussion section add the current chapters 7 and 8.

Response: We appreciate the reviewer’s suggestion regarding restructuring. However, as this manuscript is intended as a short review, we opted to maintain a concise format rather than introduce separate Results and Discussion sections. The requested elements—coffee properties, bioactive components, variability in composition, and clinical implications—are already integrated within the narrative flow of the review. We believe this approach maintains clarity and coherence while remaining consistent with the scope and word limits of a short review.

4. In the search strategy you mention that you also took into account animal studies but I can't find a table that highlights the experiments carried out on animals with active components in coffee, it is recommended that you add them.

Response: Added new Table 4 (Animal Studies of Coffee/Key Components in Periodontitis/Bone Models). This table synthesizes studies such as ligature-induced periodontitis with CGA, caffeine impacts on alveolar bone under orthodontic force, and systemic coffee/caffeine effects on bone metabolism. Additionally, we have also introduced Table 3 for in-vitro studies.

5. In figure 1 the description is very general, the exact mechanism presented in the drawing must be described.

Response: We thank the reviewer for this helpful suggestion. We have revised the Figure 1 legend to provide a more detailed mechanistic description. The updated legend now reads: “Figure 1. Schematic representation of the host–microbe interactions driving periodontal bone loss. A dysbiotic subgingival biofilm stimulates macrophages to release pro-inflammatory cytokines (IL-6, TNF-α) and matrix metalloproteinases (MMPs), which increase receptor activator of nuclear factor-κB ligand (RANKL) expression. Elevated RANKL promotes osteoclast differentiation and activity, resulting in alveolar bone resorption typical of periodontitis.” Cross-references to supporting studies are included in the main text.

6. Figure 2, it has no point, the active components can only be listed in the text.

Response: We removed Figure 2 and integrated its content into the Results text and a concise bulleted list of components with primary periodontal/bone actions in Table 2.

7. Figure 3. It does not represent anything, is it generated by A.I? The hypothesis presented in the image that you can provide polyphenolic compounds using a syringe and an extension perfusion cable is unfeasible.

Response: We removed Figure 3. To avoid any potential misinterpretation, we eliminated speculative delivery depictions and instead expanded the Discussion to consider realistic translational avenues.

8. Lines 442, 456, future research, the ideas provided should not be stated separately, they should be stated in the form of a statement.

Response: Revised according to recommendations

Reviewer 2 Report

Comments and Suggestions for Authors

Dear Authors

My recommendations

  1. In the abstract, authors must clearly state the scientific novelty of this review.
  2. Consider adding “coffee polyphenols” as keywords.
  3. Did the authors consider information on how coffee properties depend on its growing location and/or variety? Is there such scientific data?
  4. Epidemiological studies showing coffee as a risk factor (e.g., through caffeine or sugar additives) deserve slightly more weight to balance the discussion.
  5. Osteoclast-related effects are underdeveloped compared to osteoblast-focused discussion. Expanding on direct effects of coffee polyphenols on osteoclastogenesis should be added.
  6. Epidemiological studies showing coffee as a risk factor (e.g., through caffeine or sugar additives) deserve slightly more weight to balance the discussion.
  7. Figure 1. The structures should be slightly enlarged, and the names of the compounds should be aligned. All structures in 1 should also be included, for example, ferulic acid.
  8. Figures 2 and 3 are mentioned but not fully described in the text. A short caption elaborating their content (chemical structures, conceptual model) would help to readers.
  9. In Table 2, consider adding a column summarizing the net periodontal implication (protective, neutral, or harmful).
  10. Are there any studies devoted to other activities of coffee polyphenols besides those discussed in this paper? If so, this subsection could also be added (at the authors' discretion).
  11. Define acronyms upon first use in the text (e.g., CGA, MAPK, ROS) even though they are listed later.
  12. Generally clear, but some sentences are long and complex. Breaking them into shorter units would improve accessibility for a wider readership. Correct small typos (e.g., “interlukin” should be “interleukin”).
  13. References must be formatted in one style, please check them carefully and add DOI.

Major revision

Comments on the Quality of English Language

Generally clear, but some sentences are long and complex. Breaking them into shorter units would improve accessibility for a wider readership. Correct small typos (e.g., “interlukin” should be “interleukin”).

Author Response

  1. In the abstract, authors must clearly state the scientific novelty of this review. response: We thank the reviewer for this helpful suggestion. We have revised the abstract and added "The novelty of this short review lies in its integration of in vitro, animal, and epidemiologic evidence to delineate dose- and context-dependent actions of coffee polyphenols, particularly chlorogenic and ferulic acids on periodontal inflammation and alveolar bone remodeling, with special emphasis on osteoclast-related mechanisms that have not been synthesized previously."
  2. Consider adding “coffee polyphenols” as keywords. Response: Added coffee polyphenols to the Keywords. (pX)  
  3. Did the authors consider information on how coffee properties depend on its growing location and/or variety? Is there such scientific data?   Response: Yes. We added a subsection “Variability by Origin/Variety/Roast/Brew” summarizing how botanical variety, terroir, roast degree, and preparation alter polyphenol/caffeine profiles and potential biological impact, with appropriate references.   

4. Epidemiological studies showing coffee as a risk factor (e.g., through caffeine or sugar additives) deserve slightly more weight to balance the discussion.   Response: We thank the reviewer for this suggestion. The potential risks of coffee consumption, particularly when confounded by sugar and sweeteners, are already addressed in the Epidemiology section of our review (lines 224, 368, 408, 413, 409, 464 and 466). In this section, we discuss studies reporting associations between higher coffee intake and periodontitis or tooth loss, and we note how such findings are influenced by the addition of sugar/cream and by consumption patterns. We believe this adequately balances both the beneficial and potentially harmful aspects of coffee intake in relation to periodontal outcomes.   5. Osteoclast-related effects are underdeveloped compared to osteoblast-focused discussion. Expanding on direct effects of coffee polyphenols on osteoclastogenesis should be added.   Response: We appreciate the reviewer’s observation. We agree that the available evidence on the direct effects of coffee polyphenols on osteoclastogenesis is more limited compared to osteoblast-related findings. While a comprehensive expansion was not possible due to this limitation, we have highlighted this gap in the text and added references that address osteoclast-related mechanisms where available (e.g., suppression of RANKL-dependent NF-κB signaling by CGA and ferulic acid, and caffeine-enhanced osteoclastogenesis under compressive stress/orthodontic force) this also mentioned in animal study section. This provides readers with the existing evidence while also underscoring the need for further research in this area.   6. Figure 2 and 3. The structures should be slightly enlarged, and the names of the compounds should be aligned. All structures in 1 should also be included, for example, ferulic acid.   Response: We thank the reviewer for the suggestion. To streamline the manuscript and avoid redundancy with the text, we decided to remove Figure 2 and 3 entirely. The compounds and their mechanisms, including ferulic acid, are now described in the main text.   7. In Table 2, consider adding a column summarizing the net periodontal implication (protective, neutral, or harmful).   Response: Done. Table 2 now includes a “Net periodontal implication” column with a concise verdict based on each study’s primary outcome(s) and context    8. Are there any studies devoted to other activities of coffee polyphenols besides those discussed in this paper? If so, this subsection could also be added (at the authors' discretion).   Response: We thank the reviewer for this valuable suggestion. While we recognize that coffee polyphenols have been studied in other contexts (e.g., anti-glycation, antimicrobial melanoidin-related effects, microbiome modulation), the available evidence linking these activities directly to periodontal bone remodeling is limited. For this reason, we did not expand this subsection in the current manuscript, but we have noted in the Discussion that such additional activities may represent promising directions for future investigation.  

9. Define acronyms upon first use in the text (e.g., CGA, MAPK, ROS) even though they are listed later.

Response: Done. All acronyms now defined at first mention in the main text and figures. Any acronyms that are mentioned only once in the text are mentioned in the acronyms section below to avoid increasing the words for the text and to keep it concise.  

10. Generally clear, but some sentences are long and complex. Breaking them into shorter units would improve accessibility for a wider readership. Correct small typos (e.g., “interlukin” should be “interleukin”). Response: We thank the reviewer for their comments, we have changed the sentences and made them easier.

11. References must be formatted in one style, please check them carefully and add DOI. Response: We thank the reviewer for their comment. Since MDPI has a specific style of citation that they want, we have made sure to keep it standardised to Vancouver style of citations through Zotero application. For articles that were not able to get through the app, we have added the DOI for each of them. 

Round 2

Reviewer 1 Report

Comments and Suggestions for Authors

The authors have responded satisfactorily to the comments, but there are still some details that need to be revised.

1. I maintain the request that the search strategy and the method of obtaining the figures be included in a materials and methods chapter, they have no place in an introduction, that's natural.
2. In the new search strategy you mention the period 2010-2025, but in the bibliography you have studies from 2000 (line 576), 2008 (line 577), 2009 (line 617).
3. Line 434-dose of 400 mg of caffeine must be argued with a citation.
4. Lines 488-524 "future research" and "future investigations" should be reorganized as a statement, not separated into bullet points.
5. The study does not cover the less good side of caffeine, there are so many studies that show the negative effects of caffeine, for example, a paragraph should have been written about that too.

Author Response

  1. I maintain the request that the search strategy and the method of obtaining the figures be included in a materials and methods chapter, they have no place in an introduction, that's natural.

Response: We thank the reviewer for his comments, we have now added a new “Materials and Methods” section under which the Search Strategy is present. This can be found in line 104.

  1. In the new search strategy you mention the period 2010-2025, but in the bibliography you have studies from 2000 (line 576), 2008 (line 577), 2009 (line 617).

Response: We thank the reviewer for ponting this vital information out. We have now changed the timeline to January 2000, which can be found in line 113.

  1. Line 434-dose of 400 mg of caffeine must be argued with a citation.

Response: Already been cited with the article “Antonio J, Newmire DE, Stout JR, Antonio B, Gibbons M, Lowery LM, et al. Common questions and misconceptions about caffeine supplementation: what does the scientific evidence really show? J Int Soc Sports Nutr. 2024 Dec 31;21(1):2323919”

  1. Lines 488-524 "future research" and "future investigations" should be reorganized as a statement, not separated into bullet points.

Response: We thank the reviewer for pointing this out, we have now concise everything in a paragraph, which can be found in lines 504-535.

  1. The study does not cover the less good side of caffeine, there are so many studies that show the negative effects of caffeine, for example, a paragraph should have been written about that too.

Response: The article does talk about the bad effects of caffeine, however, they may not come out very well as they are spread out throughout the article. To make this better understandable, we thank the reviewer for pointing this out and hence have added a paragraph explaining the bad effects of caffeine in lines 495-502.

Reviewer 2 Report

Comments and Suggestions for Authors

Thanks to the authors for the revised manuscript. I think that the authors have adequately addressed the comments and most of my concerns. Therefore, I have no further comments. The research paper can be accepted for publication in this journal.

Author Response

Thanks to the authors for the revised manuscript. I think that the authors have adequately addressed the comments and most of my concerns. Therefore, I have no further comments. The research paper can be accepted for publication in this journal.

Response: We sincerely thank the reviewer for their time, valuable insights, and constructive feedback throughout the revision process. Your detailed comments greatly helped us refine the manuscript and improve its clarity, scientific depth, and overall quality. We truly appreciate your positive assessment and recommendation for acceptance.
